# Differences in Spinal Posture and Movement Between Adult Females with Anorexia Nervosa and Age- and Sex-Matched Normal-Weight Controls

**DOI:** 10.3390/jcm14113723

**Published:** 2025-05-26

**Authors:** Munkh-Erdene Bayartai, Gabriella Tringali, Roberta De Micheli, Adele Bondesan, Enrica Ventura, Laura Abbruzzese, Alessandro Sartorio

**Affiliations:** 1Department of Physical and Occupational Therapy, School of Nursing, Mongolian National University of Medical Sciences, Ulaanbaatar 14210, Mongolia; 2Experimental Laboratory for Auxo-Endocrinological Research, Istituto Auxologico Italiano, Istituto di Ricovero e Cura a Carattere Scientifico (IRCCS), 28824 Piancavallo-Verbania, Italy; g.tringali@auxologico.it (G.T.); r.demicheli@auxologico.it (R.D.M.); a.bondesan@auxologico.it (A.B.); sartorio@auxologico.it (A.S.); 3Division of Eating and Nutrition Disorders, Istituto Auxologico Italiano, Istituto di Ricovero e Cura a Carattere Scientifico (IRCCS), 28824 Piancavallo-Verbania, Italy; e.ventura@auxologico.it; 4Division of Auxology, Istituto Auxologico Italiano, Istituto di Ricovero e Cura a Carattere Scientifico (IRCCS), 28824 Piancavallo-Verbania, Italy; l.abbruzzese@auxologico.it

**Keywords:** anorexia nervosa, adult females, spinal mobility, spinal posture

## Abstract

**Background:** Anorexia nervosa is an eating disorder characterised by distorted eating behaviour, physical and mental health problems, and the highest mortality rate among psychiatric disorders. Although anorexia nervosa appears to be associated with alterations in the spine, studies investigating the characteristics of spinal postures and mobility in individuals with anorexia nervosa are scarce to date. The present study aims to examine the relationship between anorexia nervosa and spinal posture and mobility by comparing people with anorexia nervosa to age-matched, normal-weight controls. **Methods:** Spinal posture and mobility were evaluated using a radiation-free back scan, the Idiag M360 (Idiag, Fehraltorf, Switzerland). Between-group differences were assessed using a two-way analysis of variance. **Results:** Adult females suffering from anorexia nervosa exhibited reduced lumbar [difference between groups (Δ) = 10.5°, 95% CI 4.6°–16.5°, *p* < 0.001] and thoracic (Δ = 8.8°, 95% CI 2.4°–15.2°, *p* = 0.007) curvatures compared to normal-weight controls. The only difference observed in spinal movements between the two groups was thoracic flexion, which was greater in individuals with anorexia nervosa (Δ = 8.4°, 95% CI 2.1°–14.4°, *p* = 0.009). **Conclusions:** These findings emphasise the need to consider spinal posture and thoracic mobility in the musculoskeletal assessment of anorexia nervosa. Interventions aimed at improving spinal postures may help to develop effective rehabilitative strategies for addressing spinal problems associated with anorexia nervosa and thus contributing to counteract the possible further worsening with advancing age.

## 1. Introduction

Anorexia nervosa is an eating disorder characterised by distorted eating behaviour, physical and mental health problems, and the highest mortality rate among psychiatric disorders. It is also associated with reduced quality of life, increased healthcare costs, and lower income levels. Hospitalised individuals with anorexia nervosa were found to have a five times higher mortality risk compared to the general population [1]. Anorexia nervosa affects 4% of females and 0.3% of males, with its occurrence increasingly rising among subjects aged 15 and younger [2]. Anorexia nervosa also impacts multiple body systems, with the musculoskeletal system being one of the most commonly affected [3,4,5].

Anorexia nervosa adversely affects the muscles, bones, ligaments, tendons, and cartilage that make up the musculoskeletal system [3,4,5]. Decreased bone strength and osteoporosis, leading to a higher risk of fractures, are frequently observed in individuals with anorexia nervosa, especially in females [3,4]. While research on the relationship between anorexia nervosa and soft tissue injuries is limited compared to studies on its link to fracture risk, a recent observational study found a higher prevalence of soft tissue injuries, including cruciate ligament tears, meniscal tears, and ankle ligament sprains, in individuals with anorexia nervosa [6]. A recent systematic review reported that subjects with anorexia nervosa experience reduced muscle size and decreased energy expenditure compared to those without the disorder. However, the impact of these alterations on the physiological decline remains unclear [7]. A prospective observational study following severely malnourished individuals with anorexia nervosa, where malnutrition is prevalent, revealed that malnutrition can lead to muscle weakness, with the axial muscles group being more affected than the peripheral muscles group [5]. These authors also found that muscle weakness could improve once subjects with anorexia nervosa partially recover weight [5]. A systematic review involving randomised controlled trials reported that subjects with anorexia nervosa experienced improved muscle strength and quality of life through participation in strength training, high-intensity resistance, and stretching exercises [8]. A better understanding of musculoskeletal manifestations associated with anorexia nervosa will be essential for designing approaches that improve the health and quality of life of people with this clinical condition.

Anorexia nervosa appears to be associated with alterations in the spine [9]. A previous review study reported that people with anorexia nervosa tend to have postural misalignments, such as increased sway in the lumbar spine, anterior pelvic tilt, and spinal hyperextension [9]. The spinal manifestation in individuals with anorexia nervosa appears to be affected by the combination of decreased bone strength, low bone density, skeletal muscle weakness, and reduced body mass index [9,10]. Nevertheless, studies investigating the characteristics of spinal postures and mobility in anorexia nervosa are scarce to date. Therefore, the present study aims to examine these characteristics in individuals with anorexia nervosa through the use of a safe, reliable, non-invasive, and radiation-free device [11,12] and to compare the results with those recorded in a group of age-matched females of normal weight.

## 2. Materials and Methods

A cross-sectional design was used in the present study to assess the relationship between anorexia nervosa and spinal posture and mobility, following STROBE (Strengthening the Reporting of Observational Studies in Epidemiology) guidelines.

### 2.1. Participants

The present study included thirty-two women aged between 18.1 and 59 years (mean age + SD: 27.7 + 9.9 years) with anorexia nervosa (BMI + SD: 14.3 + 1.8 kg/m^2^). Participants were consecutively recruited from the Division of Eating and Nutrition Disorders at Istituto Auxologico Italiano, IRCCS, Piancavallo-Verbania, a tertiary care centre specialised in the rehabilitation of obesity and eating disorders in northern Italy. Participants were eligible for the study if they were female (due to the higher prevalence of eating disorders in women compared to men [1]), between 18 and 60 years old, and had a diagnosis of anorexia nervosa based on the Diagnostic and Statistical Manual of Mental Disorders, Fifth Edition (DSM-5), as determined by the Structured Clinical Interview–Clinical Version 5 (SCID-5 CV). All patients provided written informed consent before participation. Participants were excluded if they had a comorbid psychiatric disorder diagnosed according to DSM-5 criteria or any other medical condition that might affect their ability to participate in the study. Participants included in the study were at their first access to our third-level centre, with a previous history of only outpatient visits. Once enrolled, participants were asked to complete self-report questionnaires to obtain demographical and clinical data relevant to the study. Normal-weight, age-matched, female controls (mean age + SD: 27.5 + 9.3 years; BMI + SD: 22.5 + 2.2 kg/m^2^) were recruited among the hospital’s medical, research, and administrative staff and among friends and colleagues. The control group excluded participants with representative disorders, such as scoliosis, or a history of surgical interventions. Convenience sampling was utilised due to its practicality in accessing readily available populations. Although this approach inherently limits generalizability, efforts were however made to mitigate these limitations by targeting subjects of the same age and social status.

The study was approved by the Ethical Committee no. 5, Lombardy Region (registration number: 207/24; date of approval: 23 April 2024; research order code: 01C416). All procedures were conducted according to the Helsinki Declaration of 1975 and its later amendments or comparable ethical standards.

### 2.2. Measurements

The Idiag M360 scan tool (Idiag, Fehraltorf, Switzerland) assessed spinal posture, spinal mobility, and hip movements [13]. A single measurement was obtained per participant using the device to evaluate spinal posture in a static upright position and to assess spinal mobility during dynamic movements, including flexion, extension, and lateral bending. The Idiag M360 is a reliable, non-invasive, and radiation-free device designed to measure spinal posture and mobility (Figure 1). It measures the angles of each vertebral joint and the sacral slope through computer-assisted analysis. During the measurement, two rolling wheels built into the device follow the vertebral spinous processes from the seventh cervical vertebra to the third sacral vertebra. The device records the posture and mobility of each spinal segment in both the sagittal and frontal planes, with data collected at a frequency of 150 Hz [13,14]. These parameters are subsequently converted by an analogue-to-digital converter and transmitted to a computer for further analysis.

Both spinal and hip movements were analysed in the sagittal and coronal planes. Participants were asked to perform a range of motion tasks, including spinal flexion and extension and lateral bending to both sides, while maintaining an upright standing position [15]. For the assessment of spinal flexion, participants were instructed to stand upright and bend forward to the maximum extent comfortably achievable without flexing the knees. Subsequently, for spinal extension, they were instructed to extend as far as was comfortably possible, also maintaining straight knees throughout the movement. Spinal lateral bending was evaluated by instructing participants to bend sideways and slide one arm down along the corresponding leg as far as comfort allowed, while avoiding any flexion or extension movement. This assessment was conducted in accordance with protocols outlined in previous research studies for assessing spinal posture and movement [13,15,16]. The research team was trained through educational videos to evaluate spinal posture and mobility in normal and pathological conditions.

The device has been proven to be valid and reliable for assessing spinal posture and movement in a broad range of subjects, involving those with normal weight, overweight, and obesity [13,14,17,18]. Research on the validity of spinal curvature measurements using X-ray examinations demonstrated a strong correlation between radiographic measurements and those obtained with a spinal mouse [13,17]. The spinal ranges of motion reported in earlier research are closely consistent with the values measured by the Idiag M360 [14]. Additionally, a previous comparative study assessing the validity and reliability of the device in evaluating lumbar flexion found that the segmental and overall lumbar movements measured by radiography closely aligned with the values provided by the device [18]. The device demonstrated moderate to good reliability in evaluating spinal posture and mobility in non-obese individuals [14]. In this study, the average standard error of measurement was about 2°, and the intraclass correlation coefficients for inter-examiner reliability ranged from 0.63 to 0.93. A previous study evaluated the device’s reliability in obese individuals, reporting intra-rater intraclass correlation coefficients for spinal posture between 0.86 and 0.94, with standard error of measurement values ranging from 0.58° to 0.70°. For spinal mobility, the intra-rater intraclass correlation coefficients ranged from 0.57 to 0.80 in the coronal plane and from 0.87 to 0.98 in the sagittal plane [15].

Body composition was determined using a multifrequency tetrapolar bioelectrical impedance analyser (BIA, Human-IM Scan, DS-Medigroup, Milan, Italy), applying a current of 800 μA at a frequency of 50 kHz. Following the method described by Lukaski [19], the subjects rested in the supine position for 20 min before measurement, with their arms and legs relaxed and avoiding contact with other body parts. To minimise measurement errors, efforts were made to standardise the variables influencing the measurements’ validity, reproducibility, and precision. Fat mass was determined as the difference between body mass and fat-free mass.

### 2.3. Data Processing

For each spinal parameter, the range of motion was calculated in the frontal and sagittal planes by subtracting the range of motion measurements at the standing position from the end-of-motion ranges for each spinal segment. The segmental mobility was defined as the angular measurement between two adjacent spinal vertebrae during range of motion tasks. The total global ranges of motion for the spine, which include lumbar and thoracic movements, were calculated by adding the five segmental range of motion values for the lumbar spine and the twelve for the thoracic spine. The sacral tilt at the end of lumbar flexion and extension from the standing position was considered as hip flexion and extension, respectively [12,15]. The lumbar-to-hip ratio was determined by dividing the lumbar range of motion by the total of the lumbar and hip ranges of motion during trunk flexion in the sagittal plane. Pelvic tilt was defined by the angular difference between the sacral angle and the vertical axis. Pelvic incidence was calculated using the formula Pelvic tilt = 0.44 × Pelvic incidence − 11°, as established in a previous study [20].

### 2.4. Statistical Analysis

Descriptive and inferential statistical analyses were performed using R software (version 4.2.2) [21]. Descriptive statistics were used to report mean values, standard deviations (SD), the number of individuals in each group, and the percentage for factors such as participants’ age, gender, spinal posture and movement, hip motion, and lumbar-to-hip ratio. The normality of the data was evaluated using the Shapiro–Wilk test.

The independent samples *t*-test was used for data with a normal distribution, the Wilcoxon rank sum test for non-normally distributed data, and Pearson’s chi-square test for categorical variables to compare the demographic and anthropometric characteristics between the anorexia nervosa group and the normal-weight control group. A two-way analysis of variance (ANOVA), while controlling for age, was performed to evaluate significant differences in spinal posture and mobility between the two groups. Pairwise post hoc comparisons were performed using the “emmeans v 1.6.3” software package following the ANOVA analyses [22]. A *p*-value of less than 0.05 was considered statistically significant.

## 3. Results

The demographic and anthropometric characteristics of the participants are summarised in Table 1. The mean age and height were not different between individuals with anorexia nervosa and normal-weight individuals, whereas BMI and weight were remarkably lower in those with anorexia nervosa. Figure 2 presents segmental posture and movements of the spine among normal-weight controls compared to individuals with anorexia nervosa. Differences between the two groups were frequently observed in the segmental postures of the spine. For example, reduced segmental angles were more frequently observed in the lumbar vertebrae than in the thoracic vertebrae (Figure 2, Table 2).

### Relationships Between Anorexia Nervosa Characteristics and Spinal Posture and Movements

Statistically significant differences in both segmental and global spinal postures were observed between individuals with anorexia nervosa and normal-weight controls. Individuals with anorexia nervosa had reduced segmental kyphotic angles of the spine in thoracic vertebrae T3/4 and smaller segmental lordotic angles in lumbar vertebrae L2/3, L3/4 and L4/5 relative to normal-weight controls. Adult females suffering from anorexia nervosa exhibited reduced lumbar and thoracic curvatures compared to normal-weight controls. The only difference observed in spinal movements between the two groups was thoracic flexion, which was greater in individuals with anorexia nervosa. The lumbar-to-hip ratio was not different between the two groups (Table 2).

## 4. Discussion

The aim of the present study was to investigate the relationship between anorexia nervosa and spinal posture and movements in adult females with anorexia nervosa, compared to age- and sex-matched normal-weight controls. The main finding was that anorexia nervosa was associated with alterations in spinal posture and movements. Adult females suffering from anorexia nervosa exhibited reduced lumbar and thoracic curvatures compared to female individuals of normal weight. The only difference observed in spinal mobility between individuals with anorexia nervosa and those of normal weight was in thoracic flexion, which was larger in those with anorexia nervosa. No differences were observed in sacral kyphosis, pelvic incidence, and the remaining spinal mobility between individuals with anorexia nervosa and normal-weight controls. These findings imply that anorexia nervosa has a greater impact on spinal posture than on spinal mobility, highlighting the importance of taking account of spinal posture and thoracic mobility in the musculoskeletal assessment of individuals with anorexia nervosa.

Anorexia nervosa was associated with loss of thoracic and lumbar curvatures, characterised by reduced thoracic kyphosis and decreased lumbar lordosis. Anorexia nervosa appears to affect the lumbar lordosis more than the thoracic kyphosis, as reduced segmental angles were more frequently observed in the lumbar vertebrae than in the thoracic vertebrae. Although research specifically investigating characteristics of spinal posture and motion in individuals with anorexia nervosa is sparse, previous studies have reported skeletal complications associated with anorexia nervosa [3]. Individuals suffering from anorexia nervosa often experience poor bone health and an increased risk of fractures or osteoporosis due to strict lifestyle choices and prolonged nutritional restriction [3,4,6]. Previous studies also found that reduced bone mineral density at the lumbar spine frequently occurs in individuals with anorexia nervosa compared to age-matched controls [3,23,24]. The loss of thoracic and lumbar curvatures observed in the current study could potentially be a biomechanical disadvantage, attenuating shock absorption and increasing stress on vertebrae and intervertebral discs [25,26]. A previous review highlighted the importance of engaging in postural exercises for individuals with anorexia nervosa to alleviate physical symptoms and enhance self-esteem as they can experience postural alterations of the spine associated with weak muscles [27]. The loss of lumbar lordosis leads to increased pain and degenerative changes in the spine, as the paraspinal muscles in the cervical and lumbar areas predominantly maintain an upright posture [28]. Prospective studies have shown that reduced lumbar lordosis elevated the risk of developing low back pain [29], commonly experienced by individuals with anorexia nervosa [27]. The loss of lumbar lordosis also contributes to a more flattened back, leading to a greater strain on spinal muscles, such as the paraspinal muscles, which can result in pain [25]. The decreased thoracic kyphosis observed in this study may be a compensatory response to the reduction in lumbar lordosis. Therefore, approaches aimed at improving spinal postures may help to develop effective strategies for addressing spinal problems associated with anorexia nervosa.

Individuals with anorexia nervosa exhibited greater thoracic flexion compared to normal-weight controls, while no significant differences were found in other spinal movements. The literature often highlighted movement restrictions in individuals with anorexia nervosa due to muscle stiffness and weakness, although research specifically examining spinal kinematics in this population remains limited. A greater thoracic flexion observed in individuals with anorexia nervosa may be explained by a compensatory adjustment for reduced thoracic kyphosis. Reduction in thoracic and lumbar curvatures leads to spinal flattening, resulting in a posterior displacement of the trunk and necessitating increased thoracic flexion to maintain sagittal alignment [26,30]. These findings imply that the influence of anorexia nervosa may be greater on spinal posture than on spinal movements. However, further research exploring the relationships between key features of anorexia nervosa, such as muscle loss and spinal kinematics, is necessary to confirm these findings, as existing literature on this topic remains scarce.

We acknowledge that the present study has some limitations. The cross-sectional design used in this study cannot provide evidence on a causal relationship between anorexia nervosa and alterations in spinal posture and movements. The device used in this study was designed to measure spinal posture and movements only in the sagittal and frontal planes, limiting our ability to assess spinal rotation in the horizontal plane. It is important to consider that spinal movements were assessed only from the upright posture, and that differences in skeletal structure between the two groups were not controlled, which may influence the interpretation of the present findings. Pelvic incidence was estimated using the sacral angle measured by the Idiag M360 device, which is less accurate compared to pelvic incidence measurements derived from X-ray imaging, regarded as the gold standard. Furthermore, the results of the present study, conducted on a relatively small study group, will have to be confirmed with a more extensive study population, including individuals of both genders and different age ranges.

Nevertheless, the present study was strengthened by the accurate selection of two age- and sex-matched subgroups (individuals with anorexia nervosa vs. normal-weight individuals), thus providing detailed insights into the specific characteristics of the spine, including global and segmental posture and spinal mobility of adult females with anorexia nervosa.

In conclusion, anorexia nervosa was associated with alterations in spinal posture and mobility. Adult females with anorexia nervosa had reduced lumbar and thoracic curvatures compared to normal-weight individuals. The only difference observed in spinal mobility between individuals with anorexia nervosa and those of normal weight was in thoracic flexion, which was larger in those with anorexia nervosa. These findings imply that spinal posture appears to be more influenced by anorexia nervosa than spinal mobility, highlighting the importance of taking account of spinal posture and thoracic mobility in the musculoskeletal assessment of individuals with anorexia nervosa. Interventions aimed at improving spinal postures may help to develop effective rehabilitative strategies for addressing spinal problems associated with anorexia nervosa and thus contributing to counteract the possible further worsening with advancing age.

## Figures and Tables

**Figure 1 jcm-14-03723-f001:**
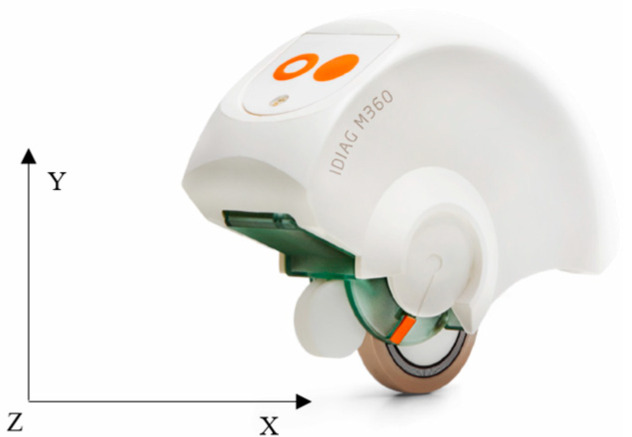
Idiag M360. Y-axis (vertical), X-axis (medial–lateral) and Z-axis (anterior–posterior) orientations.

**Figure 2 jcm-14-03723-f002:**
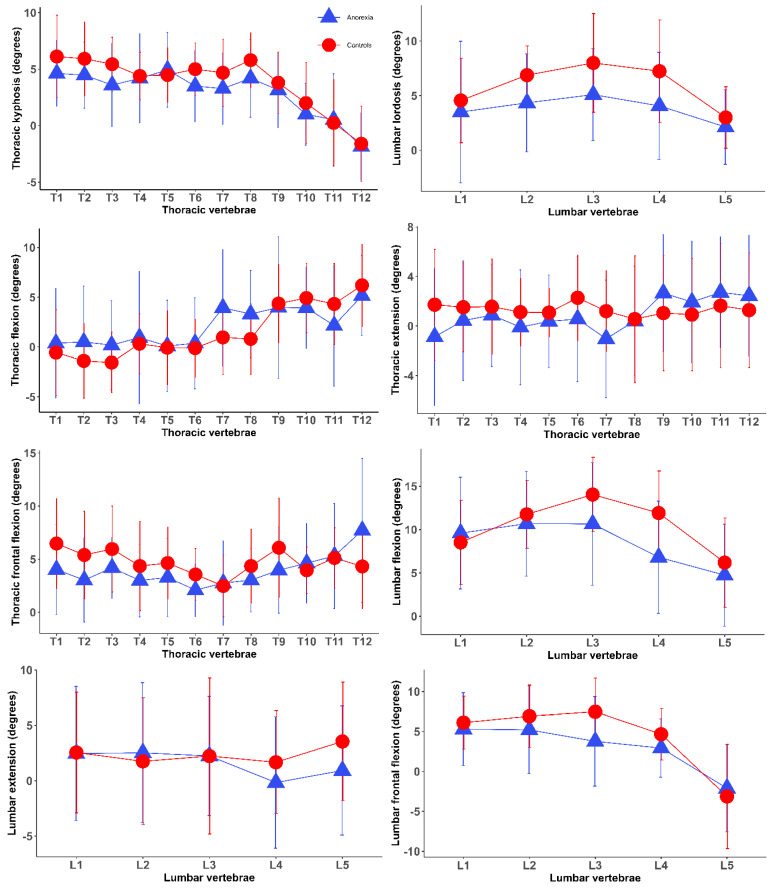
Segmental posture and movements of the spine in the group of individuals with anorexia nervosa and the group of normal-weight controls (mean and standard deviations). Positive and negative values indicate kyphosis/flexion and lordosis/extension, respectively. Controls: normal-weight controls; Anorexia: individuals with anorexia nervosa.

**Table 1 jcm-14-03723-t001:** Demographic and anthropometric parameters of the participants (mean  ±  standard deviations).

Variables	Individuals with Anorexia Nervosa (*n* = 32)	Normal-Weight Controls (*n* = 25)	*p*-Value
Age (years)	27.7 (9.9)	27.5 (9.3)	0.83 ^W^
Weight (kg)	37.1 (5.6)	62.8 (6.6)	<0.0001 ^W^
Height (cm)	161.2 (6.7)	167.0 (6.4)	0.82 ^W^
BMI (kg/m^2^)	14.3 (1.8)	22.5 (2.2)	0.002 ^W^
Fat mass (kg)	4.1 (1.9)	15.6 (5.7)	<0.0001 ^W^
Fat-free mass (%)	89.2 (4.9)	75.2 (8.9)	<0.0001 ^W^

*p*-value: the differences between the two groups, assessed using Wilcoxon’s rank sum test ^W^. BMI: body mass index.

**Table 2 jcm-14-03723-t002:** Differences in spinal postures and movements between individuals with anorexia nervosa and normal-weight controls.

Variables	Anorexia Nervosa *n* = 32		Normal-Weight Controls *n* = 25		Differences in Spinal Posture and Mobility (95% CI)	*p*-Value
	EMM	SE	EMM	SE		
Spinal postures						
Thoracic kyphosis (Th1-12)	37.5	2.1	46.3	2.3	−8.8 (−15.2 to −2.4)	0.007 *
Lumbar lordosis	19.1	1.9	29.6	2.2	−10.5 (−16.5 to −4.6)	0.0008 *
Sacral kyphosis	13.0	1.6	15.6	1.8	−2.6 (−7.5 to 2.3)	0.28
Pelvic incidence	54.5	3.6	60.4	4.1	−5.9 (−16.9 to 5.1)	0.29
Posture of each individual spinal segment						
T1/2	4.6	0.5	6.1	0.6	−1.5 (−3.3 to 0.3)	0.09
T2/3	4.5	0.5	5.9	0.6	−1.4 (−3.1 to 0.2)	0.08
T3/4	3.6	0.6	5.4	0.6	−1.8 (−3.5 to −0.1)	0.03 *
T4/5	4.2	0.6	4.4	0.6	−0.2 (−1.9 to 1.5)	0.78
T5/6	4.9	0.5	4.5	0.6	0.4 (−1.1 to 2.0)	0.55
T6/7	3.5	0.5	5.0	0.6	−1.5 (−3.0 to 0.1)	0.05
T7/8	3.3	0.5	4.7	0.6	−1.4 (−3.0 to 0.2)	0.08
T8/9	4.2	0.5	5.8	0.6	−1.6 (−3.2 to 0.01)	0.05
T9/10	3.2	0.5	3.8	0.6	−0.6 (−2.3 to 1.0)	0.44
T10/11	1.0	0.6	2.0	0.6	−1.0 (−2.7 to 0.7)	0.24
T11/12	0.5	0.7	0.2	0.8	0.3 (−1.8 to 2.4)	0.79
T12/L1	1.8	0.5	1.6	0.6	0.2 (−1.4 to 1.9)	0.79
L1/2	3.5	0.9	4.6	1.1	−1.1 (−4.0 to 1.9)	0.47
L2/3	4.4	0.7	6.9	0.8	−2.5 (−4.6 to −0.5)	0.01 *
L3/4	5.1	0.7	8.0	0.8	−2.9 (−5.2 to −0.5)	0.01 *
L4/5	4.1	0.8	7.2	0.9	−3.1 (−5.8 to −0.5)	0.02 *
L5/S1	2.1	0.5	3.0	0.6	−0.9 (−2.6 to 0.8)	0.31
Spinal mobility						
Thoracic (°)						
Flexion	26.4	2.0	18.1	2.2	8.4 (2.1 to 14.4)	0.009 *
Extension	14.4	3.2	16.0	3.6	−1.6 (−11.4 to 8.2)	0.75
Lateral flexion	41.1	6.9	61.4	6.9	−20.4 (−40.6 to −0.1)	0.37
Lumbar (°)						
Flexion	45.4	2.6	52.5	2.9	−7.1 (−15.0 to 0.8)	0.08
Extension	7.0	2.5	11.8	2.9	−4.8 (−12.4 to 2.9)	0.22
Lateral flexion	16.9	2.0	22.1	2.2	−5.2 (−11.2 to 0.9)	0.09
Hip mobility						
Hip (°)						
Flexion	44.8	3.3	49.1	3.7	−4.3 (−14.1 to 5.5)	0.38
Extension	4.3	2.1	7.8	2.4	−3.5 (−10.1 to 2.9)	0.28
Lumbar-to-hip ratio	0.5	0.02	0.5	0.03	−0.01 (−0.08 to 0.08)	0.99

*p*-value (controlling for age): the differences between the two groups. EMM (estimated marginal means): for the total range of motion values in each spinal segment across different spinal regions. SE: standard error. CI: confidence interval. * Anorexia nervosa vs. normal-weight controls *p* < 0.05. The lumbar-to-hip ratio was determined by dividing the lumbar range of motion by the total of the lumbar and hip ranges of motion during trunk flexion in the sagittal plane. “T” and “L” represent thoracic and lumbar vertebrae, respectively. The “/” is used between two adjacent vertebrae to denote segmental postures. For instance, T1/2 refers to the segmental angle formed by the first and second thoracic vertebrae.

## Data Availability

Raw data will be available on www.zenodo.org after the acceptance of the manuscript upon a reasonable request to the corresponding author.

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
