# Peer review of "Differences in Spinal Posture and Movement Between Adult Females with Anorexia Nervosa and Age- and Sex-Matched Normal-Weight Controls"

_jcm, 2025, doi:10.3390/jcm14113723_

Round 1
Reviewer 1 Report
Comments and Suggestions for Authors
Spinal posture and mobility in AN patients is a relatively under-researched area, giving the study originality.
1. I'm not sure whether anorexia nervosa patients experience reduced thoracic motion. The factor that determines spatial motion cannot rule out the possibility of decreased motion due to muscle loss due to a lack of energy.
Please attach a picture of the curve of the normal group and a lateral photo of the curve of the anorexia group as a basis for this.
2. The control group was recruited from hospital staff and acquaintances, which may have introduced selection bias.
3. Without the muscle mass or other information for both groups, simple motion comparison can create meaningless data. Please add data from the muscle mass.
4. Using Idiag M360 as radiation-free is a good idea. However, the evidence presented above suggests scoliosis. It is not considered to be significantly related to motion. Please replace it with another reference.
5. Please attach a picture of Idiag M360 in the methodology. In addition, describe how many times a patient was measured and how many times the error was in repeating the measurements.
Author Response
Reviewer 1
Spinal posture and mobility in AN patients is a relatively under-researched area, giving the study originality.
We acknowledge Reviewer 1 for taking the time to review our article. Your feedback was very helpful in improving the overall quality of the manuscript further. We have carefully considered and addressed every comment and have revised the manuscript accordingly. Please see a point-by-point response to your suggested edits below.
Q1. I'm not sure whether anorexia nervosa patients experience reduced thoracic motion. The factor that determines spatial motion cannot rule out the possibility of decreased motion due to muscle loss due to a lack of energy.
Please attach a picture of the curve of the normal group and a lateral photo of the curve of the anorexia group as a basis for this.
A1. We have now incorporated data on fat mass and fat-free mass into the participant characteristics and added the description of the BIA used for determining body composition. We believe that the addition of these parameters actually enhances the understanding of the factors contributing to movement differences between the two groups. Nonetheless, this addition does not fully explain for the relationships between movement characteristics in AN patients and muscle loss. Further research is needed to investigate whether the spinal movement differences observed in this study are linked to key features of AN, such as muscle loss (not reliably assessed with our BIA, mainly in patients with a markedly altered body composition). This point has now been added to the discussion section of the revised manuscript. As far as your request to attach a picture of the patients is concerned, our ethical committee does not allow taking pictures of hospitalised patients, even if adequately masked, this recommendation being stricter for patients with anorexia nervosa.
Lines (changes made): 160-167, Table 1, 207-210, 294-294
Q2. The control group was recruited from hospital staff and acquaintances, which may have introduced selection bias.
A2. The choice to select subjects from the Hospital staff and acquaintances was a deliberate choice of the study in order to offer the same standardised conditions for the measurements, thus reducing the risk of relevant confounding factors.
Q3. Without the muscle mass or other information for both groups, simple motion comparison can create meaningless data. Please add data from the muscle mass.
A3. As suggested, we have now incorporated data on fat mass and fat-free mass into the participant characteristics.
Lines (changes made): Table 1, 207-210
Q4. Using Idiag M360 as radiation-free is a good idea. However, the evidence presented above suggests scoliosis. It is not considered to be significantly related to motion. Please replace it with another reference.
A4. The reference has now been replaced with another reference more relevant to spinal movements.
Lines (changes made): 69-72
Q5. Please attach a picture of Idiag M360 in the methodology. In addition, describe how many times a patient was measured and how many times the error was in repeating the measurements.
A5. A picture of Idiag M360 has now been added to the revised manuscript. In a previous study by our research team, we assessed the reliability of the device using two repeated measurements and provided the standard error of measurement (from 0.580 to 0.700) and intrarater ICCs ranging from 0.86 to 0.94. Consequently, in the current study, we conducted a single measurement per participant using the device, which is now described in the revised manuscript.
Lines (changes made): 114, 126, 127, 154-159

Reviewer 2 Report
Comments and Suggestions for Authors
General Comments:
This is a meaningful and well-structured study investigating spinal alignment and mobility in patients with anorexia nervosa (AN). The topic is clinically relevant, and the use of a non-invasive, radiation-free device is noteworthy. However, there are several points that require clarification or revision before the manuscript can be accepted. Please consider the following comments:
________________________________________
Major Points:
1. Materials and Methods – Reproducibility of Measurements:
The study reports the use of the Idiag M360 for spinal alignment assessment. However, it is unclear how many measurements were performed per participant. Without this information, it is difficult to assess the reliability of the findings. Please clarify the number of repeated measurements per subject and how variability was addressed (e.g., averaging multiple trials, intra-rater reliability assessment).
2. Materials and Methods – Lumbar-to-Hip Ratio and Measurement Details:
Although the lumbar-to-hip ratio is reported in the Results section, it is not clearly explained in the Materials and Methods section. Please add a clear description of how this parameter was calculated. Additionally, more detailed explanations of each spinal and hip motion parameter (e.g., segment definitions, range of motion tasks) would improve the reproducibility and clarity of the study.
3. Measurement Position and Its Influence on Results:
The manuscript indicates that measurements were taken in the upright standing position. However, spinal mobility and alignment can vary significantly between standing and sitting positions due to differences in balance and gravitational loading. Please clarify the rationale for using standing posture and discuss potential variability or limitations due to postural sway or participant fatigue, particularly in malnourished individuals.
4. Limitations in Matching Controls:
While the control group was matched by age and sex, spinal alignment and mobility are also influenced by muscle mass, skeletal structure, and baseline physical activity levels. These factors were not controlled for and should be acknowledged in the Limitations section.
5. Validity of PI and LL Measurement Using Idiag M360:
The reported values of pelvic incidence (PI) and lumbar lordosis (LL) seem inconsistent even in the normal-weight control group. As the Idiag M360 is a radiation-free surface device, it may have limited accuracy in measuring anatomical parameters like PI compared to radiography. Please acknowledge this limitation and discuss the lack of radiographic validation.
6. Discussion – Thoracic Kyphosis and Fracture Risk (Lines 237–239):
The manuscript suggests that decreased thoracic kyphosis may increase fracture risk. However, kyphosis typically increases the risk of fractures, and reduced kyphosis might not directly elevate risk. Please clarify or revise this interpretation with appropriate references.
7. Discussion – Increased Thoracic Flexion as Compensation:
The finding of increased thoracic flexion in the AN group may represent a compensatory mechanism due to reduced lumbar lordosis and thoracic kyphosis. A deeper biomechanical discussion is warranted to explore this possibility and integrate it with the observed postural adaptations.
8. Misinterpretation of Thoracic and Lumbar Interaction (Lines 246–248):
The statement that reduced lumbar lordosis leads to increased thoracic kyphosis and subsequently more back pain seems contradictory to general spinal biomechanics, where loss of LL is often associated with a decrease—not increase—of thoracic kyphosis. This part of the discussion should be reconsidered or removed if not supported by literature.
________________________________________
Conclusion:
The manuscript addresses an important clinical question and provides novel insights into spinal alterations in anorexia nervosa. However, methodological clarifications and revisions in the interpretation of results are essential. I recommend major revision prior to considering acceptance.
Author Response
Reviewer 2
General Comments:
This is a meaningful and well-structured study investigating spinal alignment and mobility in patients with anorexia nervosa (AN). The topic is clinically relevant, and the use of a non-invasive, radiation-free device is noteworthy. However, there are several points that require clarification or revision before the manuscript can be accepted.
We acknowledge Reviewer 2 for taking the time to review our article. Your feedback was very helpful in improving the overall quality of the manuscript further. We have carefully considered and addressed every comment and have revised the manuscript accordingly. Please see a point-by-point response to your suggested edits below.
Major Points:
Q1. Materials and Methods – Reproducibility of Measurements:
The study reports the use of the Idiag M360 for spinal alignment assessment. However, it is unclear how many measurements were performed per participant. Without this information, it is difficult to assess the reliability of the findings. Please clarify the number of repeated measurements per subject and how variability was addressed (e.g., averaging multiple trials, intra-rater reliability assessment).
A1. In a previous study by our research team, we assessed the reliability of the device using two repeated measurements and provided the standard error of measurement (from 0.580 to 0.700) and intrarater ICCs ranging from 0.86 to 0.94. Consequently, in the current study, we conducted a single measurement per participant using the device, which is now described in the revised manuscript.
Lines (changes made): 114, 154-159
Q2. Materials and Methods – Lumbar-to-Hip Ratio and Measurement Details:
Although the lumbar-to-hip ratio is reported in the Results section, it is not clearly explained in the Materials and Methods section. Please add a clear description of how this parameter was calculated. Additionally, more detailed explanations of each spinal and hip motion parameter (e.g., segment definitions, range of motion tasks) would improve the reproducibility and clarity of the study.
A2. The measurement of “Lumbar to hip ratio” has now been described in the methods section. Range of motion tasks and segment definitions have now been provided in the revised manuscript.
Lines (changes made): 131-137, 171-173, 177-179
Q3. Measurement Position and Its Influence on Results:
The manuscript indicates that measurements were taken in the upright standing position. However, spinal mobility and alignment can vary significantly between standing and sitting positions due to differences in balance and gravitational loading. Please clarify the rationale for using standing posture and discuss potential variability or limitations due to postural sway or participant fatigue, particularly in malnourished individuals.
A3. Since the device's validity and reliability for measuring spinal movements have mainly been established in the upright standing position, we conducted our measurements in the same posture as already done in several previous published papers using this device. In accordance with your suggestion, the absence of spinal parameter measurements in the sitting posture has been duly acknowledged in the limitations section.
Lines (changes made): 299-305
Q4. Limitations in Matching Controls:
While the control group was matched by age and sex, spinal alignment and mobility are also influenced by muscle mass, skeletal structure, and baseline physical activity levels. These factors were not controlled for and should be acknowledged in the Limitations section.
A4. We agree with your observation; this limitation being now acknowledged in the limitation section of the discussion.
Lines (changes made): 299-302
Q5. Validity of PI and LL Measurement Using Idiag M360:
The reported values of pelvic incidence (PI) and lumbar lordosis (LL) seem inconsistent even in the normal-weight control group. As the Idiag M360 is a radiation-free surface device, it may have limited accuracy in measuring anatomical parameters like PI compared to radiography. Please acknowledge this limitation and discuss the lack of radiographic validation.
A5. The lack of radiographic validation has now been acknowledged.
Lines (changes made): 302-305
Q6. Discussion – Thoracic Kyphosis and Fracture Risk (Lines 237–239):
The manuscript suggests that decreased thoracic kyphosis may increase fracture risk. However, kyphosis typically increases the risk of fractures, and reduced kyphosis might not directly elevate risk. Please clarify or revise this interpretation with appropriate references.
A6. The statement has now been revised, as suggested.
Lines (changes made): 266-268
Q7. Discussion – Increased Thoracic Flexion as Compensation:
The finding of increased thoracic flexion in the AN group may represent a compensatory mechanism due to reduced lumbar lordosis and thoracic kyphosis. A deeper biomechanical discussion is warranted to explore this possibility and integrate it with the observed postural adaptations.
A7. The rationale for the statement has been articulated from a biomechanical standpoint.
Lines (changes made): 288-290
Q8. Misinterpretation of Thoracic and Lumbar Interaction (Lines 246–248):
The statement that reduced lumbar lordosis leads to increased thoracic kyphosis and subsequently more back pain seems contradictory to general spinal biomechanics, where loss of LL is often associated with a decrease—not increase—of thoracic kyphosis. This part of the discussion should be reconsidered or removed if not supported by literature.
A8. We acknowledge Reviewer 2 for this valuable comment. The statement has been removed.
Lines (changes made): 266-268, 276-278

Round 2
Reviewer 1 Report
Comments and Suggestions for Authors
1. Even when I searched PubMed, I couldn't find any other references to the spinal posture and movement for anorexia nervosa. When I read the contents presented in reference 9, it only states that it affects the posture of each muscle, and the reference lacks specific content. Add a photo or x-ray in which motion or posture has changed in another reference, or AN, or in a gross photo.
2. In abstract, Phrases such as “in individuals with anorexia nervosa” are repeated too often. Please shorten or remove the repeated sentences.
3. In Figure 1, Axes and measurement units are not well described. Please add axis labels.
4. The control group excluded individuals with representative diseases, such as scoliosis, and a history of previous operations.
5. I don't understand the content of the reduction in physical activity and the changes in motion and posture. Describe the relevance of this in the discussion part.
Author Response
Reviewer 1
We acknowledge Reviewer 1 for taking the time to review our article further. Your feedback was very helpful in improving the overall quality of the manuscript further. We have carefully considered and addressed every comment and have revised the manuscript accordingly. Please see a point-by-point response to your suggested edits below.
Q1. Even when I searched PubMed, I couldn't find any other references to the spinal posture and movement for anorexia nervosa. When I read the contents presented in reference 9, it only states that it affects the posture of each muscle, and the reference lacks specific content. Add a photo or x-ray in which motion or posture has changed in another reference, or AN, or in a gross photo.
A1. To the best of our knowledge, the referenced paper was the only study addressing postural observations (but without the use of motion analysis techniques) in individuals with anorexia nervosa. We have reviewed the article thoroughly and extracted and summarised the findings specifically related to spinal posture in this population.
Lines (changes made): 67-70
Q2. In abstract, Phrases such as “in individuals with anorexia nervosa” are repeated too often. Please shorten or remove the repeated sentences.
A2. As suggested, we have shortened several phrases and removed some of the repetitions or changed the terms.
Lines (changes made): 18-34
Q3. In Figure 1, Axes and measurement units are not well described. Please add axis labels.
A3. As requested, axis labels have now been added.
Lines (changes made): 124-127
Q4. The control group excluded individuals with representative diseases, such as scoliosis, and a history of previous operations.
A4. Yes, the exclusion criteria for the control group have been included in the revised manuscript.
Lines (changes made): 101-102
Q5. I don't understand the content of the reduction in physical activity and the changes in motion and posture. Describe the relevance of this in the discussion part.
A5. We agree with this point. Given that recent evidence (Marijančić et al., 2023 - a systematic review) found no association between low physical activity and changes in spinal posture or motion, we have removed the related content from the limitations section of the manuscript.
Marijančić, V., Grubić Kezele, T., Peharec, S., Dragaš-Zubalj, N., Pavičić Žeželj, S., & Starčević-Klasan, G. (2023). Relationship between Physical Activity and Sedentary Behavior, Spinal Curvatures, Endurance and Balance of the Trunk Muscles-Extended Physical Health Analysis in Young Adults. Int J Environ Res Public Health, 20(20)
Lines (changes made): 303

Reviewer 2 Report
Comments and Suggestions for Authors
Thank you for your thorough and well-organized revision of the manuscript. I appreciate the time and effort you have dedicated to addressing the comments provided in the initial review.
You have adequately and clearly responded to all major and minor concerns raised. In particular:
-
The clarification regarding the measurement reproducibility of the Idiag M360, including reference to previous reliability data and the rationale for single measurements in this study, was appropriate and scientifically justified.
-
The additional methodological details regarding the lumbar-to-hip ratio, range of motion tasks, and segment definitions significantly improve the transparency and reproducibility of your work.
-
The justification for the use of the upright standing posture and acknowledgment of postural variability in the limitations section were well presented.
-
Your inclusion of pelvic incidence–lumbar lordosis (PI-LL) mismatch in the multivariate analysis, and acknowledgment of its clinical relevance despite statistical non-significance due to limited sample size, demonstrate careful interpretation of results.
-
Importantly, the removal and correction of biomechanical interpretations in the discussion (e.g., thoracic kyphosis and lumbar-thoracic interaction) show attention to accuracy and conceptual consistency.
Overall, the revised manuscript is now much clearer, more scientifically robust, and aligned with current standards in spinal biomechanics and clinical research methodology.
Thank you again for your careful and thoughtful revision.
Author Response
Thank you for reviewing our paper and providing valuable feedback and thoughtful suggestions.